# Effect of 3-D depth structure, element size, and area containing elements on total-element overestimation phenomenon

**Yusuke Matsuda**[1]*, **Saori Aida**[2], **Koichi Shimono**[3]

**1** Department of Applied Information Engineering, Faculty of Engineering, Suwa University of Science, Chino, Nagano, Japan, **2** Graduate School of Sciences and Technology for Innovation, Yamaguchi University, Ube, Yamaguchi, Japan, **3** Department of Logistics and Information Engineering, Tokyo University of Marine Science and Technology, Koto-ku, Tokyo, Japan

* matsuda_yusuke@rs.sus.ac.jp

**Data Availability Statement:** All relevant data are within the manuscript and its Supporting information files.

## Abstract

The number of elements distributed in a three-dimensional stimulus is overestimated compared to a two-dimensional stimulus when both stimuli have the same number of elements. We examined the effect of the properties of a three-dimensional stimulus (the number of overlapping stereo surfaces, size of the elements, and size of the area containing elements, on the overestimation phenomenon in four experiments. The two stimuli were presented side-by-side with the same diameters. Observers judged which of the three-dimensional standard and two-dimensional comparison had more elements. The results showed that (a) the overestimation phenomenon occurred for the three-dimensional standard stimuli, (b) the size of the areas affected the amount of overestimation, while the number of overlapping stereo surfaces and size of elements did not, and (c) the amount of overestimation increased when the stimuli included more than 100 elements. Implications of these findings were discussed in the framework of back-surface bias, occlusion, and disparity-processing interference models.

## Introduction

Humans can estimate the number of a large set of elements rather accurately. This capability is called numerosity estimation, which has been the subject of many studies. Most previous studies have presented elements on a flat surface with zero disparity (e.g., [1–14]). Their results showed that many factors affect numerosity estimation. For example, the number of elements was judged to be more numerous when the area containing elements is relatively large [2, 8, 10, 12 (cf. 13)] and when the size of elements is relatively small [5, 10]. Furthermore, the luminance of elements [9], location of retinal stimulation [11], and connectedness of the elements [4, 6, 8] have also been found to affect numerosity judgments. These studies suggest that several factors should be considered to explain numerosity estimation [14].

Recently, four studies have indicated that a depth structure affects numerosity estimation. Schütz [15] reported a back-surface-element overestimation phenomenon, where the number of elements on the back surface of a three-dimensional (3-D) stimulus depicting two motion-

**Funding:** This work was partly supported by Grants-in-Aid for Scientific Research (https://www.jsps.go.jp/english/e-grants/), Grant-in-Aid for Young Scientists(19K20645) for YM; Grant-in-Aid for Young Scientists(B)(17K18187) for SA; Grant-in-Aid for Young Scientists(21K18027) for SA; Grant-in-Aid for Scientific Research(B)(23330215, 15H03463) for KS. The funders had no role in study design, data collection and analysis, decision to publish, or preparation of manuscript.

**Competing interests:** The authors have declared that no competing interests exist.

transparent planes was overestimated compared to that on a front surface. Aida et al. [16], Aida et al. [17], and Aida [18] found a total-element overestimation phenomenon. The perceived number of static elements on a 3-D stimulus was higher than that on a two-dimensional (2-D) flat stimulus when each contained the same number of elements. In Aida et al. [16], Aida et al. [17], and Aida [18], a 3-D stimulus depicted parallel, overlapping, transparent stereoscopic surfaces (POTS) or two non-overlapping surfaces horizontally or vertically separated at different depths (stepwise surfaces). Thus, the four studies suggest that numerosity is not perceived independently from the depth structure of the stimuli. Accordingly, to fully understand the mechanism of numerosity estimation, it is necessary to examine numerosity estimation when the elements are presented in a 3-D stimulus as well as a 2-D stimulus.

To explain the overestimation of elements embedded in a depth structure, researchers have proposed three models: back-surface bias [15–17], occlusion [16, 17], and disparity-processing interference [17] models. However, each model has its advantages and limits. The back-surface bias model assumes that when two (or more) overlapping surfaces are perceived in-depth, a process assigns the elements in the back surface to their surrounding blank areas [19] and that the number of elements on the back surface increases [15]. The occlusion model assumes that, when observers perceive the overlapped stereo surfaces, the number of unseen elements entirely impeded by the elements on the front surface would be added [16, 17]. Notably, the total-element overestimation phenomenon can occur even when there is no background in the depth structure [17]; this challenges the two models. Finally, the disparity-processing interference model assumes that processing the disparity of 3-D stimulus elements interferes with numerosity processing and leads to the overestimation of the perceived number of elements. Nevertheless, the model can hardly explain why the elements on a back surface are overestimated compared to those on a front surface when observers perceive the overlapped stereo surface [17]. Accordingly, these models cannot account for every result concerning the overestimated numerosity for a 3-D stimulus.

Bell et al. [20] reported that observers judged the number of elements distributed in a 3-D volume to be the same as that in a 2-D stimulus. That is, the total-element overestimation phenomenon was not found in their study. Bell et al. [20] used a 3-D stimulus containing elements with seven different disparities, and when observers fused the stimulus, they perceived it as a cylindrical volume with no overlapping surfaces. Accordingly, Bell et al. [20] claimed that the overestimation phenomenon could be observed for the overlapping stereo surfaces but not for the cylindrical volume. However, Aida et al. [17] found that the overestimation phenomenon did not depend on whether they perceived overlapping surfaces or cylindrical volume. The finding suggests that factors other than the depth structure of the 3-D stimulus might have affected the overestimation phenomenon.

In this study we examine what stimulus factors contribute to the total-element overestimation phenomenon. Although there are several possible candidates for the stimulus factors, we focused on four variables of 2-D and 3-D stimuli that might contribute to the phenomenon: the number of "physical" surfaces of a 3-D stimulus, the size of elements, the size of the area containing the elements, and the number of contained elements.

We conducted four experiments to examine the stimulus factors. In Experiment 1, we re-examined the claim that the occurrence of the overestimation phenomenon depends on the depth structure of a 3-D stimulus; we used a 3-D stimulus that would be perceived as overlapping surfaces in-depth or a cylindrical volume. In Experiments 2 and 3, we manipulated the sizes of an element presented on overlapping stereo surfaces and that of an area containing elements. We selected the variables because the study that reported no overestimation phenomenon [20] used relatively small elements and areas than those reporting the phenomenon [15,

16]. In Experiment 4, we manipulated the number of elements on the surface to confirm the third experiment's finding under a broader range of element numbers.

## General methods

The studies involving human participants were reviewed and approved by the Ethical Committee for Human-Subject Research at the Tokyo University of Marine Science and Technology (28–002). We recruited human observers for this study from May 25, 2016, to March 31, 2018. The observers provided written informed consent to participate in this study.

### Observers

Thirty-five observers from a university community participated in this study. Experiment 1 was conducted with ten observers (three men) aged 18–37, Experiment 2 with ten observers (nine men) aged 20–62, Experiment 3 with twenty-three observers (ten men) aged 18–45, and Experiment 4 with fifteen observers (six men) aged 19–23. The same fifteen observers participated in Experiments 3 and 4. Two of them also participated in Experiment 1. One observer of Experiment 1 participated in Experiment 2, two observers of Experiment 1 participated in Experiment 3, and one observer of Experiment 2 participated in Experiment 3. All observers had normal or corrected-to-normal vision. The observers had normal stereo vision as measured by the Stereo Fly Test (Stereo Optical Co., Inc.).

### Apparatus

We used a Windows OS 7 computer running Matlab (R2014b) with the Psychophysics Toolbox Version 3 (PTB-3) [21, 22] for stimulus presentation. The stimuli were stereograms consisting of two half-images, each of which was presented side-by-side on a 21.5-inch monitor (GW 2265, BenQ, resolution of 1920 × 1080 pixels, refresh rate of 60 Hz). A double-mirror stereoscope was used to generate a stereo scene, and the optical distance was 63.0 cm, measured with a laser distance measurer (GLM 50 Professional, Bosch, Inc., Stuttgart, Germany) (see Fig 1A).

### Stimuli

As illustrated in Fig 1B, we used two sets of random-element stimuli: a 2-D stimulus as a comparison stimulus and a 3-D stimulus as a standard stimulus. Each standard and comparison stimulus was composed of black square elements on a gray background. In each stimulus, black elements were allocated to a circular area. The position of each element in each stimulus was randomly assigned in a circular area to avoid overlap. The element position of a standard stimulus was also manipulated to ensure binocular fusion (see [23, 24]). The sizes of the elements and circular area diameter for the standard stimulus were the same as those for the comparison stimulus. The luminances of the element and background, measured with a luminance meter (LS-100, Konika Minolta, Inc., Tokyo, Japan), were 0.62 cd/m$^2$ and 47.5 cd/m$^2$, respectively.

The standard and comparison stimuli elements had some disparities and zero disparity, respectively, concerning each experiment's monitor plane. There were two standard stimuli: a 3-D two-POTS [16–18, 25] and a 3-D volume. The elements of the first had two different disparities and would be perceived as consisting of two overlapping surfaces when fused, each parallel to the front parallel plane and at a different depth. The elements of the second had seven different disparities and would be perceived as a cylindrical cluster perpendicular to the front parallel plane when fused. The comparison stimulus would be perceived as a flat surface

(A) (B)

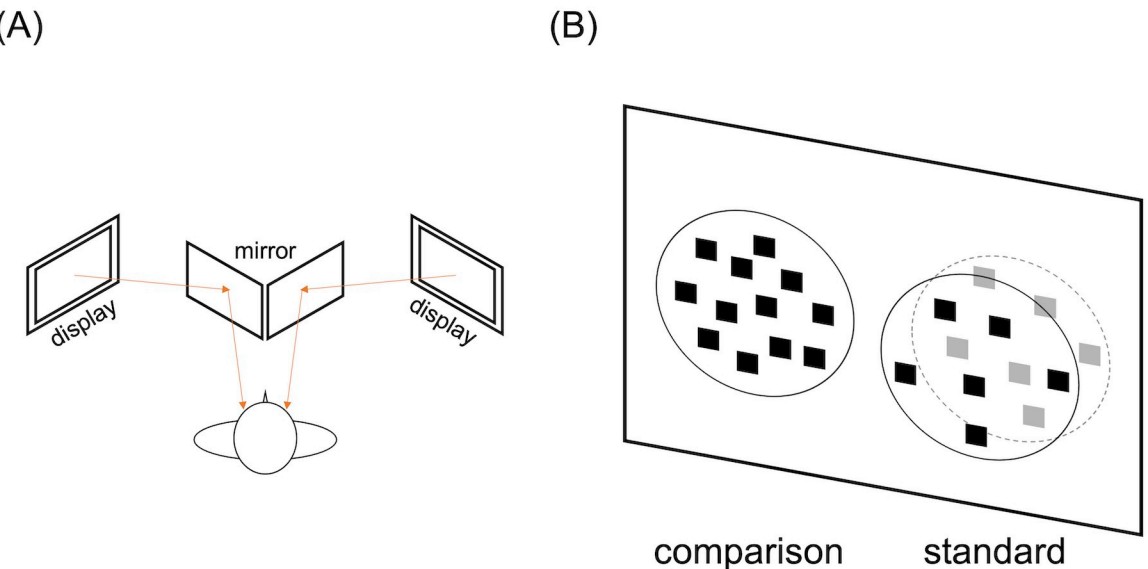

**Fig 1. Schematic illustrations of the stimulus and experimental setup.** (A) A schematic top view of the experimental setup. The setup consisted of two displays to stimulate the two eyes independently. (B) A front view of the stimulus that consisted of a 2-D comparison stimulus and a 3-D standard stimulus. While the 3-D standard is illustrated as if it consisted of a front surface with black elements and a back surface with white elements for descriptive purposes, the elements used in both surfaces were black as in the 2-D comparison.

on the monitor plane when fused. In Experiment 1, both the 3-D two-POTS and the 3-D volume stimuli were used as the standard. Experiments 2–4 used only the two-POTS stimulus as the standard.

The number of elements was fixed for the standard stimulus but was varied for the comparison stimulus in each experiment. The number of elements of the standard stimulus was 100 each in Experiments 1–3 and 50 or 150 in Experiment 4. The number of elements in the comparison stimulus was calculated for each trial for each experiment using the following equation:

$$N_{2D} = (1 + ER)N_{3D}, \tag{1}$$

where $N_{3D}$ and $N_{2D}$ are the element numbers for the standard and comparison stimuli, respectively, and $ER$ (Elements Ratio) is the ratio of the element number of the standard stimulus to that of the comparison. The procedure to set the ER values is explained in the Procedure subsection below.

## Procedure

In Experiments 1–4, observers performed a numerosity-discrimination task. They judged which of the two stimuli, a left or right stimulus, appeared more numerous and responded by pressing the left or right key on the keyboard. We instructed observers not to count the elements but to respond as soon as possible. We did not set a limitation for viewing time because the presentation would allow observers to take enough time to detect the binocular disparity and perceive the stereoscopic depth (e.g., [26]). Although the viewing time was not measured for each trial, the average total time for observers suggested that each trial should have taken a few seconds. We interpreted this viewing time as indicating that observers did not count the stimuli elements. While observers viewed the stimuli, their heads were fixed on a head-and-

chin rest. After pressing the key, the observers pressed the space key to proceed to the subsequent trial. The observers were allowed to take breaks whenever they wanted.

In each experiment, observers were asked to repeat the same session if their performance did not meet our criterion. Our criterion was the value of the coefficient of determination ($R^2$) of the psychometric function fitted to each observer's data. When the value was greater than 0.8, we analyzed the data further; when the value was less than 0.8, we discarded the observer's data and asked them to perform the same session again on a different day within three weeks, if available. Eventually, the observers whose $R^2$ value reached our criterion had a maximum of three sessions. The data of those who did not pass the criterion three times in a row or were unavailable were eliminated from the analysis. Please refer to the Data analysis sub-section below for the details of the data analysis.

## Data analysis

We calculated the ER in each experiment, at which the element number of the comparison stimulus was perceived to be the same as that of the standard stimulus. We defined the ER as the point of subjective equality ratio (PSE ratio). We calculated the PSE ratio from a psychometric function fitted to the responses, at which the number of comparison elements was perceived to be larger than that of the standard elements as a function of the ER. The psychometric function was logistic as follows:

$$y = \frac{1}{1 + exp\{-A(x - B)\}},\tag{2}$$

where $x$ is the independent value (i.e., the ER), $y$ is the dependent value (i.e., the ratio of choosing the comparison stimulus), $A$ represents the slope of the middle part of the curve, and $B$ represents the abscissa at the central point of the logistic function. The logistic function was fitted with KaleidaGraph version 4.5 (Hulinks Inc. Tokyo, Japan).

Before calculating the PSE ratio, we evaluated the goodness-of-fit of the logistic function using the $R^2$ for each condition and observer. As previously mentioned, observers might perform the same session three times depending on the $R^2$ value. In each of the three sessions, the number of comparison elements was manipulated by varying the ER values in Eq (1). In the first session, we set the ER value from -0.3 to +0.3 with an 0.1 increment step size; in the second, we manipulated the ER values with the same increment step size; and in the third, we manipulated both the ER values and increment step size (0.2). These manipulations were made to obtain an $R^2$ value of more than 0.8 after observing the previous session's data for each observer.

We present an instance of a psychometric function in Fig 2 to illustrate the method to calculate the PSE ratio. In this instance, the $R^2$ value was 0.96 and reached our criterion. We determined that when the PSE ratio was positive, the number of elements was perceived to be more numerous for the standard than for the comparison and, when negative, less numerous.

## Experiment 1

In this experiment, we manipulated the number of "physical" surfaces of 3-D stimuli to examine whether the perceived overlay of the stereo surfaces is a critical factor in the overestimation phenomenon. Bell et al. [20] and Aida et al. [16] claimed that the overestimation phenomenon could be due to the perception of the overlapping stereo surfaces. To examine the claim's validity, we used two different 3-D stimuli: physically, one depicted two surfaces (3-D two-POTS stimulus) and the other seven surfaces (3-D volume stimulus). Our manipulation of the number of physical surfaces was based on a previous study, which found that a 3-D stimulus that

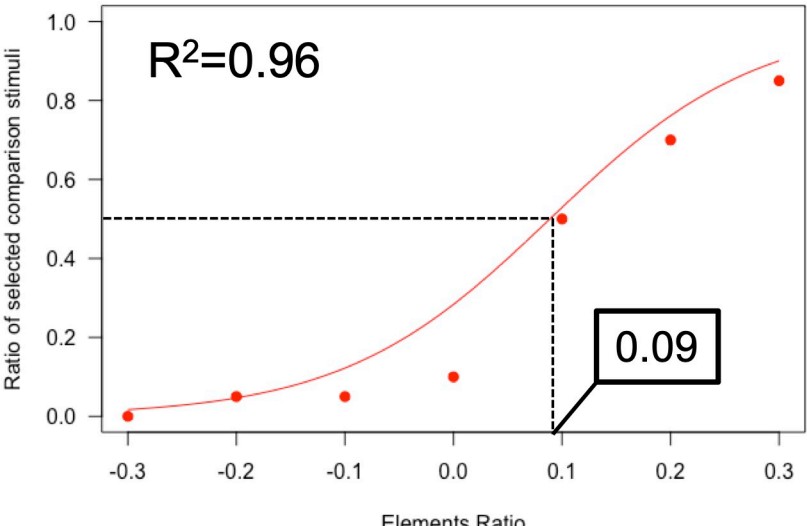

**Fig 2. Example of the psychometric function for one observer.** The red circles indicate the ratios of one observer's responses that judged the number of 2-D (comparison) stimulus elements to be larger than that of 3-D (standard) stimulus elements in each element ratio condition. The ratios were those obtained in Experiment 1. The selected ratios of comparison ($y$-axis) were plotted against the ERs ($x$-axis). The red curve indicates the best-fitted curve of the logistic psychometric function ($R^2 = 0.96$). The $x$-axis value yielding a 0.5 response ratio in the function represents the PSE ratio of the 2-D stimulus, which is indicated by the dotted vertical line. The PSE ratio value shows that the number of the comparison elements appears to be the same as that of the standard elements. In this case, the value was 0.09. If the number of standard stimulus elements is 100 (which is the same as in Experiment 1), a PSE of 0.09 indicates that "the 100 elements of the standard stimulus and the 109 elements of the comparison stimulus are perceived as the same number".

simulates five or more overlapping "stereo-surfaces" is difficult to perceive as surfaces [19]. Before the main experiment, we confirmed that all observers perceived two overlapping surfaces in-depth and one cylindrical volume for the two-POTS and the volume stimuli, respectively.

The claim has been examined by Aida et al. [17] using a 3-D five-POTS stimulus, which physically depicted five surfaces in-depth. In Experiment 2 of Aida et al. [17], the five-POTS stimulus did not necessarily provide constant perception among observers; four out of seven observers reported a volume, and the remaining three reported three overlapping surfaces. Furthermore, Aida et al. [17] reported the overestimation phenomenon irrespective of whether overlapping surfaces were perceived. Accordingly, the report suggests that the perception of overlapping surfaces is not critical in the overestimation phenomenon. Our experiment aimed to test the suggestion using a stimulus that would provide constant 3-D volumetric perception.

## Methods

**Stimuli.** The size of each element and the diameter of the area containing elements were $6.7 \times 6.7$ arcmin and 8.9 arc deg, respectively, each for the standard and comparison stimuli. The distance between the center of the standard and comparison stimuli was 13.8 arc deg. The standards were 3-D two-POTS and 3-D volume stimuli, each with 100 elements; the total element density was 1.6 elements/arc deg$^2$. For the two-POTS standard, 50 elements had +7.8 arcmin crossed disparity, and the other half had -8.2 arcmin uncrossed disparity concerning the monitor plane. For the volume standard, 14 elements had -8.2, -5.4, and -2.7 arcmin

uncrossed, 0.0 or +2.6 arcmin crossed disparity concerning the monitor plane, and among the remaining, 15 elements had +5.2 or +7.8 arcmin crossed disparity each.

The number of comparison elements was determined using Eq (1) and could differ among the number of times in each observer's session (see the Data analysis sub-section in the General methods section). In the first session, the number of comparison elements varied from 70 to 130 with an increment step size of 10 for every observer; the total element density varied from 1.1 to 2.1 elements/arc deg$^2$. In the second, the number of comparison elements varied from $70 + 100 \times \alpha$ to $130 + 100 \times \alpha$ with an increment step size of 10, where $\alpha$ is a shift ratio of the number of comparison stimuli. The value of $\alpha$ was separately selected from 0.0 to 0.3 for individual observers. In the third, the number of comparison elements varied from $40 + 100 \times \alpha$ to $160 + 100 \times \alpha$ with an increment step size of 20; the value of $\alpha$ was separately selected from 0.0 to 0.4 for individual observers. For these two sessions, the density range of the comparison elements differed among observers.

**Procedure.** Observers performed one session and might have repeated it three times. In the session, the type of 3-D standard stimulus (a two-POTS or a volume) and the number of elements in the 2-D comparison stimulus were randomly selected with 20 repetitions. Consequently, there were 280 trials (two 3-D standard × seven numbers of elements × 20 repetitions) for each observer. The placement of the 3-D standard stimulus (to the left or right of the mid-sagittal plane for each trial and observer) was randomly determined.

We calculated the $R^2$ value of the logistic function fitted to the result of the session for each observer. We found that all 10 observers passed our criterion. Then, we computed the PSE ratio. The mean number of sessions, where the ratio was calculated, across the observers and its SD were 2.2 and 0.4, respectively.

Before the first session, observers performed a practice session where the type of 3-D stimulus and the number of elements of the comparison stimulus were randomly assigned for each trial and observer. The practice session had 14 trials (two types of 3-D stimuli × seven numbers of elements in the comparison stimulus), where the location of the 3-D stimulus was randomly selected from two locations (left and right). The number of elements in the comparison was the same as used in the first session. The observers were asked whether they perceived overlaid surfaces or a 3-D volume before the numerosity-discrimination task.

## Results and discussion

First, we conducted a paired $t$-test to compare the PSE ratio between the 3-D two-POTS and 3-D volume stimuli and found that the difference in the PSE ratio was not statistically significant, $t(9) = 0.48$, $p = .64$, Cohen's $d = 0.15$. The result is shown in the left panel of Fig 3, where the mean PSE ratio averaged across 10 observers is depicted separately for each 3-D stimulus. The mean PSE ratios were 0.13 (95% CI, 0.06–0.19) and 0.12 (95% CI, 0.05–0.18) in the 3-D two-POTS and 3-D volume stimuli, respectively. The mean PSE ratio indicates that the number of elements in the 2-D comparison, perceived to have the same number as that in the 3-D two-POTS or 3-D volume stimulus, was 112 and 113, respectively. Notably, the 3-D stimulus had 100 elements.

Second, we conducted a one-sample $t$-test for the mean PSE ratio. The Benjamini-Hochberg (BH) False Discovery Rate method was used to correct $p$-values for the test [27]. The result showed that the values of the PSE ratios are significantly higher than zero: 3-D two-POTS: $t(9) = 3.94$, BH-adjusted $p = .006$, Cohen's $d = 1.25$; and 3-D volume: $t(9) = 3.61$, BH-adjusted $p = .006$, Cohen's $d = 1.14$. There was an overestimated number of elements for each 3-D stimulus (see the Data analysis sub-section in General methods). The $t$-test revealed that the overestimation phenomenon occurred and that the amount of the phenomenon was nearly

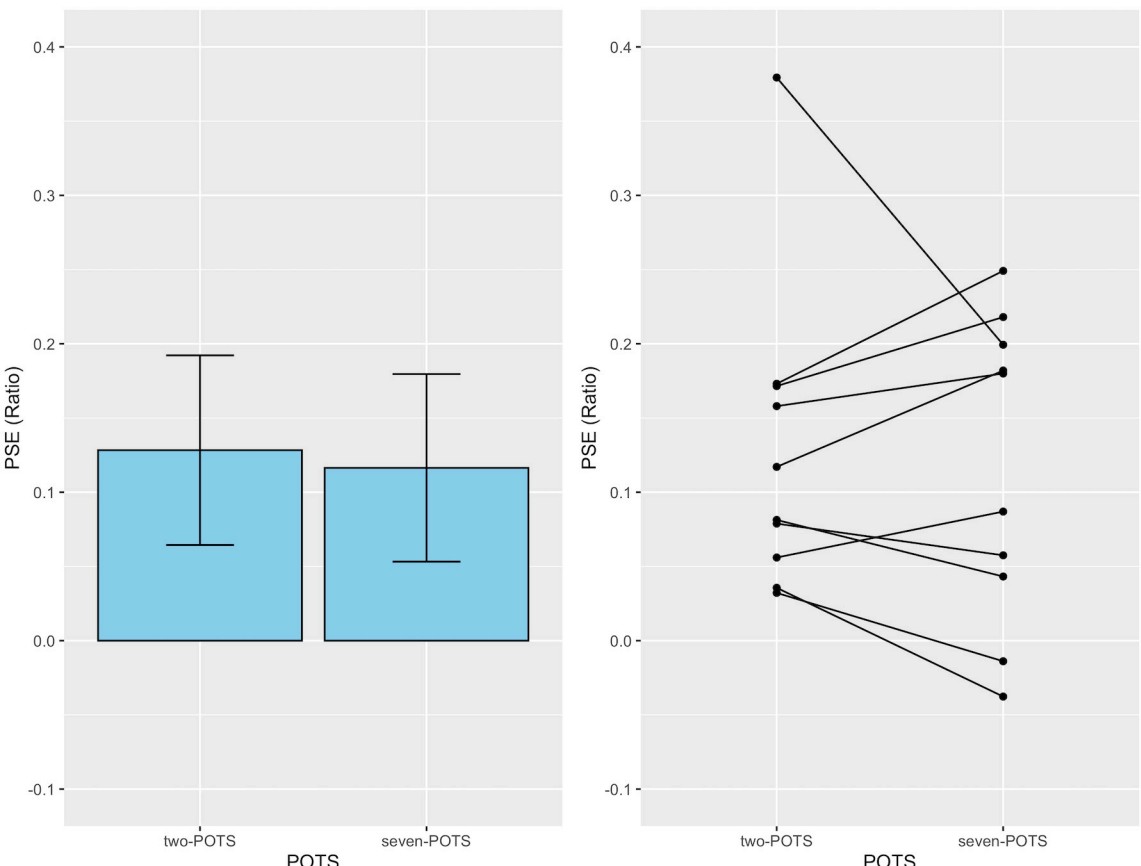

**Fig 3. Results from Experiment 1.** The left panel shows the mean PSE ratios for two types of 3-D stimuli; the left and right bars indicate the mean for the 3-D two-POTS and 3-D volume stimuli, respectively. Each error bar attached to the bar represents a 95% confidence interval (CI). The right panel shows each individual's PSE ratio for the two-POTS and volume stimuli. The black circles connected by black lines indicate data from the same observers.

the same irrespective of whether a 3-D stimulus had overlaid surfaces. This conclusion is consistent with the right panel of Fig 3, which depicts each observer's PSE ratio for the two 3-D stimuli. In the panel, the ratios of the same observer were connected by a line. The panel shows that every observer's PSE ratio was more than zero except for two observers for the 3-D volume stimulus, suggesting that most observers reported the overestimation phenomenon under the study conditions. The panel also shows that the PSE ratio was larger in the 3-D two-POTS stimulus than in the 3-D volume stimulus for five observers but reversed for the remaining five. Thus, the difference in the 3-D stimulus used had no apparent effect on the phenomenon.

However, it is necessary to show that the present results are not attributable to the procedure in Experiment 1 in order to claim that the overestimation phenomenon is specific to a 3-D stimulus. If we assumed that the observers tended to select a standard stimulus but not a more numerous stimulus in this experiment (where a standard had a different stimulus property from that of a comparison like three-dimensionality), the present results would be explainable. To examine whether the tendency existed, we conducted three control experiments where we used the 2-D standard and comparison stimuli and manipulated the shape and color of the standard elements. In the first one, the standard and comparison elements had the same attributes as those in Experiment 1; namely, each stimulus was composed of

black squares. In the second and third ones, the standard rectangular elements were red and triangular, respectively, while the other attributes of the standard and the comparison were the same as in the first. The procedure of each control experiment was essentially the same as that of the main session in Experiment 1, but the number of trials was halved (140 trials: seven ERs × 20 repetitions). Eleven observers (three men; eight women) who were aged 20–23 years conducted the experiments, each consisting of one session, where the number of comparison elements varied from 40 to 160 with an increment step size of 20. The observers performed the experiments sequentially from the first to the third control and took a break between them.

The result of each control experiment indicates that a total-element overestimation did not occur for the 2-D standard. The mean values of the PSE ratio were -0.01 (95% CI, -0.03 to 0.01), 0.02 (95% CI, -0.002 to 0.04), and 0.05 (95% CI, 0.01 to 0.08) in the control experiments 1, 2 and 3, respectively. The values were statistically not different from zero ($t$ (10) = 0.85, $p$ = .41, Cohen's $d$ = 0.26, $t$ (10) = 0.85, $p$ = .42, Cohen's $d$ = 0.26, $t$ (10) = 1.23, $p$ = .25, Cohen's $d$ = 0.37). The result indicates that no numerosity overestimation was observed for a given standard when the standard and comparison were presented side-by-side and that the standard had a property that was the same as or different from that of the comparison. Thus, the results are consistent with our claim that the overestimation phenomenon observed in this experiment is attributable to a 3-D stimulus but not to our procedure. Since the procedures used in Experiments 2–4 were essentially the same as those in Experiment 1, we decided not to run the control experiment for the following three experiments.

The results of Experiment 1 do not support the claim of Bell et al. [20] and Aida et al. [16] that the overestimation phenomenon is obtained when a 3-D stimulus is perceived to be overlapping with stereo surfaces but not obtained when it is perceived as a cylindrical volume. Instead, the results support Aida et al.'s [17] finding that the overestimation phenomenon for a 3-D stimulus is observed irrespective of whether it is perceived as overlaid surfaces or volume. If this is the case, why did Bell et al. [20] not find the phenomenon when observers perceived it as a cylindrical volume? One explanation is that instead of the overlapped stereo surface, other stimulus properties, such as the size of the elements or the area containing elements, may affect the phenomenon. In Experiments 2 and 3, we examined the effects of the size of the elements and the diameter of the areas containing elements, respectively.

## Experiment 2

In this experiment, we manipulated the size of the elements of a 3-D stimulus to determine whether it is a factor in the overestimation phenomenon. This manipulation was performed based on the observation that the size was smaller in Bell et al. [20], who did not report the phenomenon, than in Aida et al. [16], Aida et al. [17], and Schütz [15], who reported the phenomenon. The size of the element used in Bell et al. [20] was 3.0 × 3.0 arcmin, and that used in Aida et al. [17] and Aida [18], Schütz [15], and Aida et al. [16] was 5.7 × 5.7 arcmin, 6.8 × 12.0, and 8.4 × 8.4, respectively. However, notably, the smallest size of elements used in this experiment was slightly larger than that used in Bell et al. [20]. We used that size because our preliminary observation showed that when the element size was close to that used in Bell et al. [20], several observers reported difficulty or inability to perceive depth for a 3-D stimulus. We selected the smallest element size that could be perceived in-depth in this experiment because it was a prerequisite for the present experiment that the stimulus should be fused and perceived in depth.

## Methods

**Stimuli.** The element size of the 3-D standard and 2-D comparison stimuli was 4.0 × 4.0, 6.7 × 6.7, and 9.3 × 9.3 arcmin; these magnitudes will be called small, medium, or large, respectively. As in Experiment 1, the diameter of the area containing elements was 8.9 arc deg, and the center-to-center distance between the standard and comparison stimuli was 13.8 arc deg. The standard was a 3-D two-POTS stimulus with 100 elements; 50 elements had crossed (+7.8 arcmin) and 50 elements uncrossed (-8.2 arcmin) disparities, respectively, concerning the monitor plane. The total element density of the standard was 1.6 elements/arc deg$^2$, as in Experiment 1. The number of the comparison elements was determined as in Experiment 1. The total element density varied from 1.1 to 2.1 elements/arc deg$^2$ in the first session. In the second and third sessions, the $\alpha$ value was consistently zero.

**Procedure.** Observers performed one session and might have repeated it three times, as in Experiment 1. In the session, the sizes of elements (small, medium, or large) and the number of elements in the comparison stimulus were randomly selected with 10 repetitions. Consequently, there were 210 trials (three sizes of elements × seven ERs × ten repetitions) for each observer. The location of the 3-D standard was randomly determined, as in Experiment 1.

As in Experiment 1, we calculated the $R^2$ value of the logistic function fitted to the result of the session for each condition for each observer. We found that eight out of ten observers had passed the criterion. Then, we computed the PSE ratio for each observer. The eight observers' mean number of sessions and SD were 1.5 and 0.9, respectively.

Before the first session, observers performed a practice session where the size and number of elements in the comparison were randomly assigned for each trial and observer. The practice session had 21 trials (three sizes of elements × seven numbers of elements in the comparison stimulus), where the location of the standard stimulus was selected randomly from its two locations (left and right). The number of elements in the comparison was the same as those used in the first session.

## Results and discussion

First, we conducted a repeated-measures one-way ANOVA on the PSE ratio with three sizes of elements and found no significant main effect, $F(2, 14) = 0.01$, $p = .99$, *partial $\eta^2$* = 0.002. The result is shown in the left panel of Fig 4, in which the PSE ratios averaged across eight observers are depicted as a function of the size of the elements; the mean PSE ratios were 0.13 (95% CI, 0.04–0.22), 0.13 (0.04–0.22), and 0.13 (0.07–0.20) when the element sizes were small, medium, and large, respectively. The mean PSE ratios indicate that the number of elements in the 2-D comparison, judged to be the same as that in the 3-D standard with 100 elements, was 113, the same as that among three different element conditions.

Second, we conducted a one-sample *t*-test for the mean PSE ratio. The results showed that the values of PSE ratios are significantly higher than zero: small: $t(7) = 2.79$, BH-adjusted $p = .027$, Cohen's $d = 0.99$; medium: $t(7) = 2.97$, BH-adjusted $p = .032$, Cohen's $d = 1.05$; and large: $t(7) = 3.88$, BH-adjusted $p = .020$, Cohen's $d = 1.37$. The results indicated an overestimation of the number of elements in each of the three element conditions (see the Data analysis sub-section in General methods). The ANOVA and *t*-test revealed that the overestimation phenomenon was observed for each of the three element conditions. Furthermore, the size of the element did not affect the phenomenon.

The same conclusion can also be drawn from the right panel of Fig 4, which shows each observer's PSE ratio for the three element conditions. In the panel, the ratios of the same observer are connected by a line. The panel shows that every observer's PSE ratio was more than zero except for one in the small condition. The panel also indicates no apparent tendency

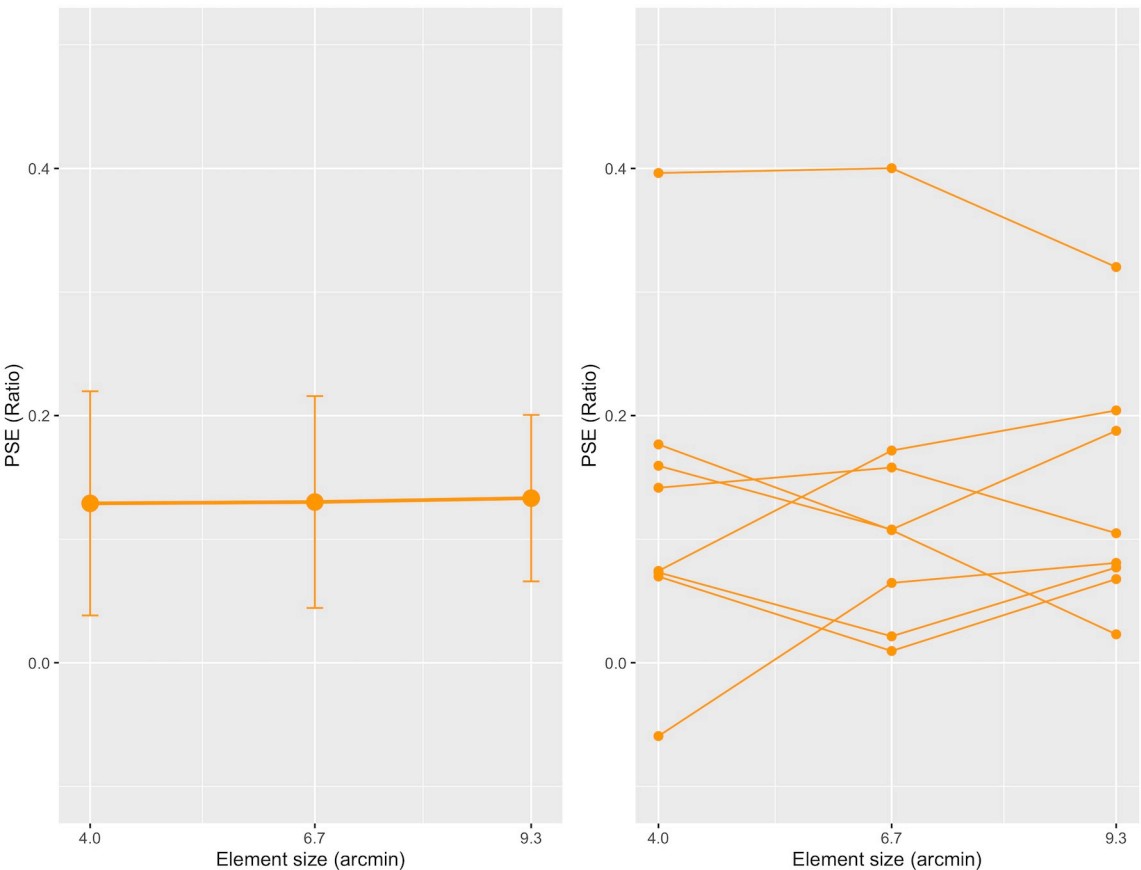

**Fig 4. Results from Experiment 2.** The left panel shows the mean PSE ratio as a function of the element size of the 3-D two-POTS stimulus. Each error bar attached to the circle represents a 95% confidence interval (CI). The right panel shows each individual's PSE ratio as a function of the element size. The orange circles connected by orange lines indicate data from the same observers.

to change the PSE ratio as a function of the element size. These results are consistent with the above conclusion that the element sizes used in this experiment do not modify the overestimation phenomenon. If the element size does not affect the overestimation phenomenon, why was the phenomenon not observed in Bell et al. [20]? We discuss this question in the General Discussion section.

We found no element size effect on the overestimation phenomenon, as discussed above. However, this finding does not mean that the element size has no effect on the perceived number of elements in a 3-D stimulus. Since we presented a 3-D and 2-D stimulus side-by-side, the finding indicates that the degree of element size effect is similar between the two stimuli. Regarding the 2-D numerosity judgment, the element size affects the numerosity judgments in its accuracy [5, 10] and precision [10, 14].

## Experiment 3

In this experiment, we manipulated the area containing elements. This manipulation was made based on the observation that the area in Bell et al. [20], who did not report the phenomenon, was smaller than in Aida [18], Aida et al. [16, 17], Schütz [15], who reported the phenomenon, except for one area in Aida [18]. The area was 31.4 arc deg$^2$ in Bell et al. [20] and it

was 21.3 and 37.2 arc deg$^2$ in Aida [18], 47.5 arc deg$^2$ in Experiment 2 and 175.3 arc deg$^2$ in Experiments 1, 3, and 4 in Aida et al. [16], 74.0 arc deg$^2$ in Aida et al. [17], and 400 arc deg$^2$ in Schütz [15]. These results can be understood if the phenomenon is less likely for a relatively small area. To examine this possibility, we varied the area from 15.5 to 138.1 arc deg$^2$, covering the range where the overestimation could or could not occur.

## Methods

**Stimuli.** The size of each element was 6.7 × 6.7 arcmins and the size of the area containing elements was 15.5, 61.7, or 138.1 arc deg$^2$ (diameter of the area: 4.4, 8.9, or 13.3 arc deg); these magnitudes will be referred to as small, medium, or large, respectively. The center-to-center distance between the standard and comparison stimuli was the same for all three size conditions, i.e., was 13.8 arc deg, as in Experiments 1 and 2. The standard and comparison stimuli were a two-POTS stimulus and a flat stimulus, respectively, as in Experiment 2. The standard had 100 elements; 50 elements had crossed (+7.8 arcmin) and 50 elements had uncrossed (-8.2 arcmin) disparities concerning the monitor plane as in Experiments 1 and 2. The total density of the standard was 6.5, 1.6, or 0.7 elements/arc deg$^2$, respectively, for the small, medium, or large area.

The number of comparison elements was determined as in Experiments 1 and 2. In the first session, the density of the comparison stimulus varied from 4.5 to 8.4, 1.1 to 2.1. or 0.5 to 0.9 elements/arc deg$^2$, respectively, for the small, medium, or large areas. In the second and third sessions, the density range of the comparison differed among observers depending on the value of $\alpha$ chosen among observers. The value was selected from 0.0 to 0.3 and 0.0 to 0.4 in the second and third sessions, respectively.

**Procedure.** Observers performed one session and might have repeated it three times, as in Experiments 1 and 2. In the session, the size of the areas (small, medium, or large) and the number of elements in the comparison stimulus were randomly selected with ten repetitions. Consequently, there were 210 trials (three areas × seven numbers of elements × ten repetitions) for each observer. The location of the 3-D standard was randomly determined, as in Experiments 1 and 2.

As in Experiments 1 and 2, we calculated the $R^2$ value of the logistic function fitted to the result of the session for each size condition for each observer. We found that 17 out of 23 observers passed the criterion. We then computed the PSE ratio for each of the 17 observers. The 17 observers' mean number of sessions and SD were 2.24 and 0.83, respectively.

Before the first session, observers performed the practice session, where the size of areas and number of elements in the comparison were randomly assigned for each trial and observer. The practice session had 21 trials (three sizes of area × seven numbers of elements), where the location of the standard stimulus was randomly selected from its two locations (left and right). The number of elements in the comparison was the same as used in the first session.

## Results and discussion

First, we conducted a repeated-measures one-way ANOVA on the PSE ratio with three different areas and found a significant main effect, $F$ (2, 32) = 4.48, $p$ = .019, partial $\eta^2$ = 0.22. A post-hoc analysis (Holm's method) showed that the differences in PSE ratios between the small and medium conditions ($t$ (16) = 2.39, adj. $p$ = .046) and between the small and large conditions were statistically significant ($t$ (16) = 2.75, adj. $p$ = .029), whereas that between the medium and large conditions was not ($t$ (16) = 0.36, adj. $p$ = .72). These results can be seen in the left panel of Fig 5 in which we depict the mean PSE ratios among 17 observers as a function

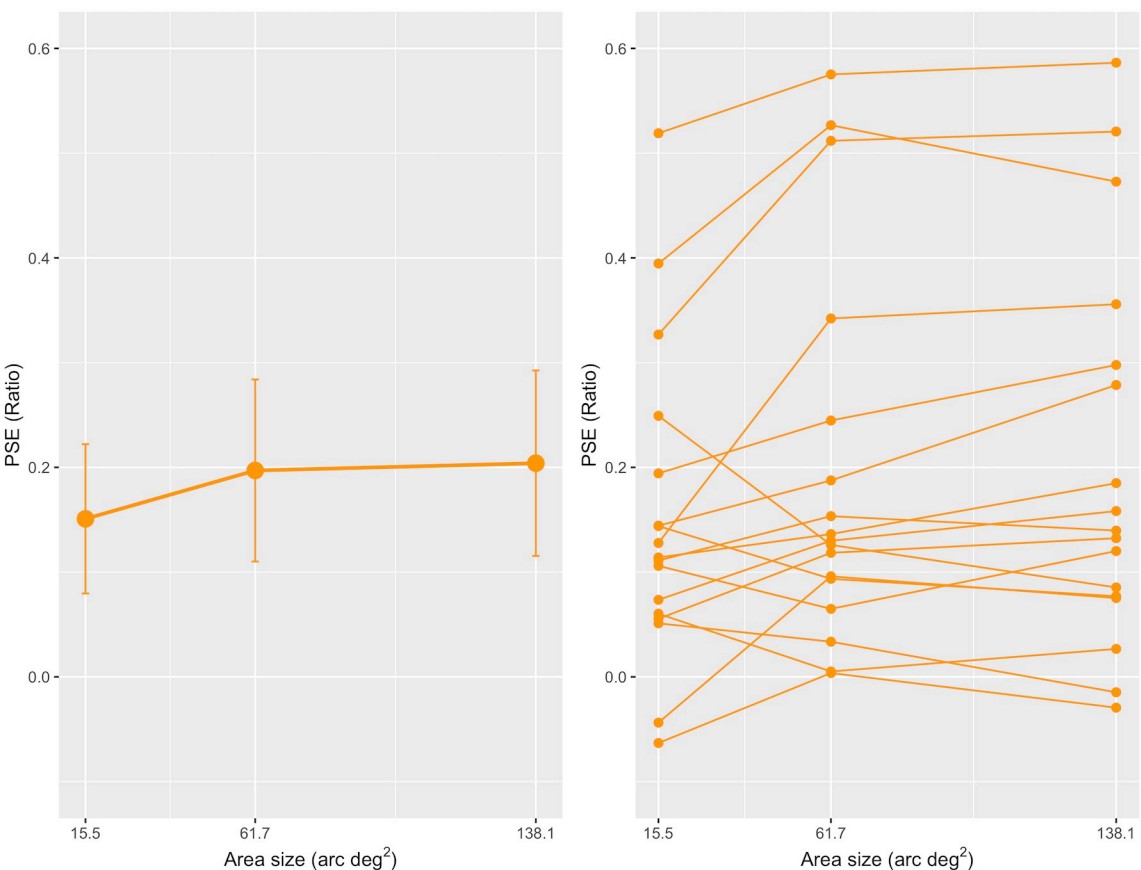

**Fig 5. Results from Experiment 3.** The left panel shows the mean PSE ratio as a function of the area (arc deg$^2$) of the 3-D two-POTS stimulus. Each error bar attached to the circle represents a 95% confidence interval (CI). The right panel shows each individual's PSE ratio as a function of the area. The orange circles connected by orange lines indicate data from the same observers.

of the diameter of the area; the mean PSE ratios were 0.15 (95% CI, 0.08–0.22), 0.20 (0.11–0.28), and 0.20 (0.12–0.29) for the small, medium, and large areas, respectively. The mean PSE ratios indicate that the number of elements in a 3-D stimulus, judged to be the same as those in a 2-D stimulus which contains 100 elements, corresponds to 115, 120, and 120 elements for the small, medium, and large conditions, respectively.

Second, we conducted a one-sample *t*-test for the mean PSE ratio and found that the PSE ratios are significantly higher than zero: the small condition: $t(16) = 4.15$, BH-adjusted $p < .001$, Cohen's $d = 1.01$; the medium condition: $t(16) = 4.44$, BH-adjusted $p < .001$, Cohen's $d = 1.08$; and the large condition: $t(16) = 4.51$, BH-adjusted $p < .001$, Cohen's $d = 1.10$. The results indicated an overestimation of the number of elements in each area condition (see General methods). The ANOVA and *t*-test showed the overestimation phenomenon for the three areas used. Further, the area affected the amount of overestimation; the amount was larger for the medium/large than the small.

Each observer's data set corresponded well to the group data: the right panel of Fig 5 shows each observer's PSE ratio as a function of the area; a line connects the ratios of the same observer. The panel shows that every observer's PSE ratio was more than zero except for two observers in the small condition and two in the large condition. The panel also shows that the PSE ratio is higher in the medium condition for 12 out of 17 observers or in the large condition

for 13 out of 17 observers than in the small condition. These results are consistent with the above conclusion.

The present results show that the overestimation phenomenon occurred for a wide range of areas (from 15.5 to 138.1 arc deg$^2$). The results are in the same line with the previous results reported the phenomenon within the range of area from 21.2 to 175.3 arc deg$^2$ [15–18 (cf. 20)]. Thus, in the literature, the overestimation phenomenon for a 3-D stimulus is relatively common even for a small area.

The present results also show that the area affected the amount of overestimation. The results appear to be inconsistent with those in Aida [18], in which the size of the area did not affect its amount. Note, however, that the range of area and the element numbers differed between this experiment and Aida's [18]. In this experiment, the effect of area emerged between the small area (15.5 arc deg$^2$) and middle area (61.7 arc deg$^2$) or large area (138.1 arc deg$^2$). In Aida [18], there was no effect between the two areas (21.2 and 37.2 arc deg$^2$). Thus, the apparent difference between the two results can be due to the different range of the area used. We addressed this issue further in the following experiment.

The present results showing the effect of area also suggest that the effect can be more prominent for 3-D than 2-D stimuli. In this study, we presented the stimuli side-by-side. For the 2-D numerosity estimation, the number of elements in a 2-D stimulus appears numerous as the area increases [2, 8, 10, 12 (cf. 13)]. Thus, the fact that the amount of overestimation for 3-D elements is larger for the smaller size than the medium or large size suggests that the effect increases more rapidly for a 3-D stimulus than a 2-D stimulus.

## Experiment 4

In this experiment, we examined the effect of area on the overestimation phenomenon under more general conditions. As discussed above, there is a difference between the result from Experiment 3 and Aida's [18] experiment concerning the area size effect on the phenomenon. The difference can be due to the difference in the element number used between the two experiments. In Experiment 3, the element number was 100, while in Aida [18], the number was relatively small (12 to 60). Thus, the area size effect would appear for a relatively large number of elements but not for a relatively small number. We examined the idea by manipulating the element number within the same area range in Experiment 3.

### Methods

**Stimuli.**   The size of each element in the standard or comparison stimulus was 6.7 × 6.7 arcmin, and the area was 15.5, 61.7, or 138.1 arc deg$^2$ (diameter of the area: 4.4, 8.9, or 13.3 arc deg); these magnitudes will be referred to as small, medium, or large, respectively. The center-to-center distance between the standard and comparison stimuli was 13.8 arc deg in all three size conditions, as in Experiment 3. The standard was a 3-D two-POTS, as in Experiments 2 and 3. Each standard surface contained half the total elements; the total number was 50 or 150. The density of the standard with 50 elements was 3.3, 0.8, or 0.4 elements/arc deg$^2$, respectively, for the small, medium, or large area, and with 150 elements was 9.7, 2.4, or 1.1 elements/arc deg$^2$, respectively, for the small, medium, or large area. The disparities of elements in the standard were the same as in Experiments 2 and 3.

The number of the comparison elements was determined as in Experiments 1–3. In the first session, the number of elements varied from 35 to 65 with an increment step size of 5 for the standard with 50 elements, and from 105 to 195 with an increment step size of 15 for that with 150 elements for each area condition. The density varied from 2.3 to 4.2, 0.6 to 1.1, and 0.3 to 0.5 elements/arc deg$^2$, respectively, for the small, medium, or large area for the standard with

50 elements, and from 6.8 to 13.6, 1.7 to 3.2, and 0.8 to 1.4 elements/arc deg$^2$, respectively, for the small, medium, or large area for the standard with 150 elements. In the second, the number of comparison elements for the standard with 50 and 150 elements varied from $35 + 50 \times \alpha$ to $65 + 50 \times \alpha$ and $105 + 150 \times \alpha$ to $195 + 150 \times \alpha$, respectively, with an increment step size of 10. In the third, the step size was set to 10 and 30 for the standard with 50 and 150 elements, respectively. As in the second, the density range of the comparison differed among observers depending on the chosen value of $\alpha$. For the standard with 50 elements, the α value was selected from 0.0 to 0.4 and 0.0 to 0.5 in the second and third sessions, respectively, and for the standard with 150 elements, the value was selected from 0.0 to 0.3 in the second session, while it was zero in the third session.

**Procedure.** Observers performed two sessions: one for the standard with 50 elements and the other for that with 150 elements. The order of the two sessions was counterbalanced. In each session, the size of areas and number of elements in the comparison stimulus were randomly selected with ten repetitions, as in Experiment 3; there were 210 trials (three sizes of areas × seven numbers of elements × ten repetitions) for each observer. The location of the 3-D standard was randomly determined using the same method as in Experiments 1–3. Similar to Experiments 1–3, the observer may have performed each session three times. Before the sessions, the observer performed a practice session whose trials were identical to those in Experiment 3 except for the area.

As in Experiments 1–3 we calculated the $R^2$ value of the logistic function fitted to the session result and found that 12 out of 15 observers met the criterion for both the standard with 50 and 150 elements, respectively. Thus, we calculated PSE ratios for the 12 observers. The mean sessions and SD for the observers were 2.67 (0.47) and 2.75 (0.43) for the 50- and 150-element standard, respectively.

## Results and discussion

First, we conducted a repeated-measures two-way ANOVA on the value of the PSE ratio with the three different areas (small, medium, and large) and two different numbers of elements in the standard stimulus (50 and 150 elements). We found a significant main effect of the number of elements, $F(1, 11) = 5.12$, $p = .045$, *partial* $\eta^2 = 0.32$, and a significant interaction, $F(2, 22) = 6.08$, $p < .008$, *partial* $\eta^2 = 0.36$. There was no significant main effect of the areas, $F(1, 11) = 1.70$, $p = .205$, *partial* $\eta^2 = 0.13$. A *post-hoc* analysis (Holm's method) showed that the difference in the PSE ratio between the small and large areas was statistically significant for the 150-element standard, $t(11) = 3.13$, adj. $p = .046$. No significant differences were found between any pair of areas for the 50-element standard (adj.$ps > .05$).

The results can be seen in Fig 6, in which the mean PSE ratio across twelve observers is depicted as a function of the area with the number of elements as the parameter. The mean PSE ratios are relatively constant among the three different areas for the 50-element condition; they were 0.18 (95% CI, 0.11–0.25), 0.18 (95% CI, 0.08–0.28), and 0.15 (95% CI, 0.03–0.27) for the small, medium, and large conditions, respectively. The mean PSE ratios indicate that the number of elements in the 2-D comparison, judged to have the same number of elements in the 3-D standard stimulus with 50 elements, corresponds to 59, 59, and 58 elements for the small, medium, and large conditions, respectively. The mean PSE ratios were higher for the medium or large area than the small area for 150-element conditions; they were 0.12 (95% CI; 0.05–0.19), 0.28 (95% CI, 0.15–0.40), and 0.30 (95% CI, 0.11–0.49) for the small, medium, and large conditions, respectively. The number of elements in the 2-D comparison, judged to have the same number of elements in the standard with 150 elements, corresponds to 168, 192, and 195 elements on average for the small, medium, and large conditions, respectively.

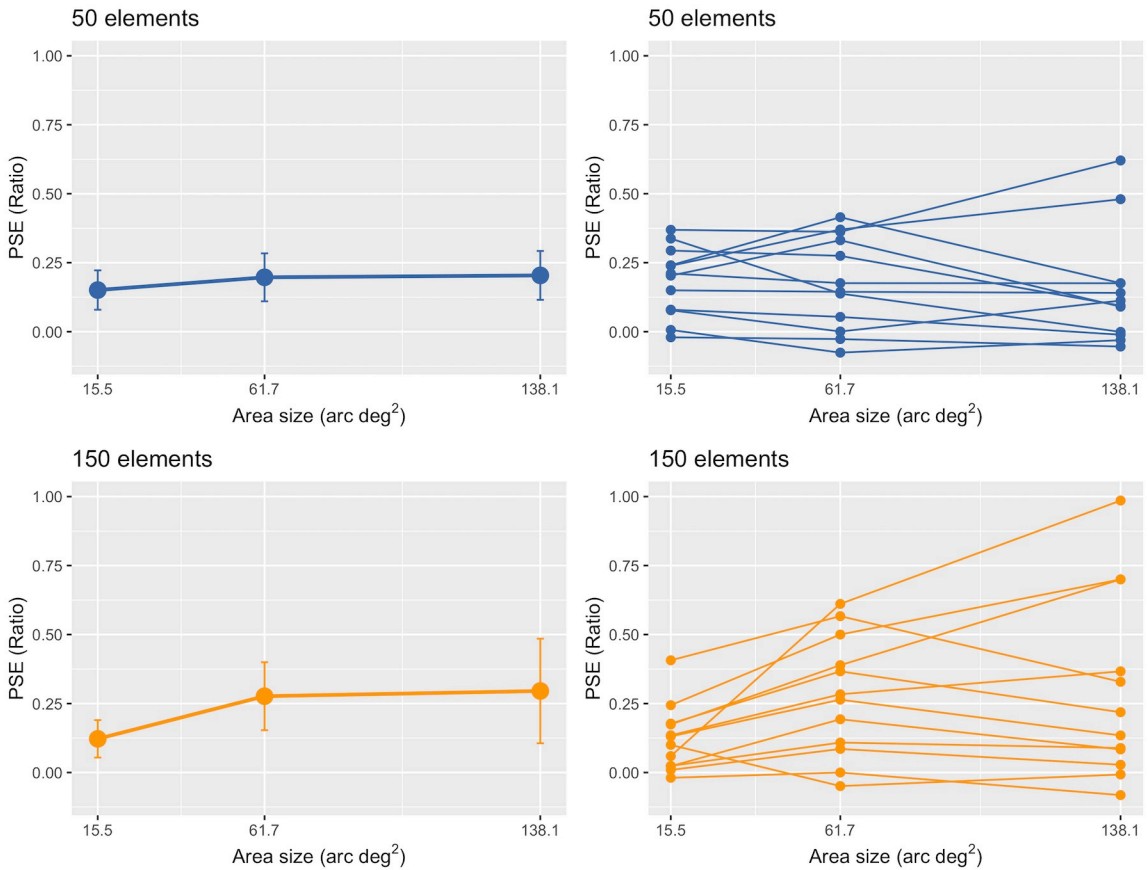

**Fig 6. Results from Experiment 4.** The left two panels show the mean PSE ratio as a function of the area (arc deg$^2$) of the 3-D two-POTS stimulus: the upper (blue circles) and lower (orange circles) panels show the data for the stimulus containing 50 and 150 elements, respectively. Each error bar attached to the circle represents a 95% confidence interval (CI). The two right panels show each individual's PSE ratio. The upper and lower panels for the stimulus contain 50 and 150 elements, respectively. The circles connected by lines indicate data from the same observers.

Second, we conducted the one-sample $t$-test for the mean PSE ratio. The result showed that the PSE ratios are significantly higher than zero for the 50-element standard: small size: $t$ (11) = 5.02, BH-adjusted $p < .001$, Cohen's $d$ = 1.45; medium size: $t$ (11) = 3.69, BH-adjusted $p$ = .008, Cohen's $d$ = 1.06; and large size: $t$ (11) = 2.53, BH-adjusted $p$ = .03, Cohen's $d$ = 0.73. For the 150-element standard, the result showed the following: small size: $t$ (11) = 3.52, BH-adjusted $p$ = .008, Cohen's $d$ = 1.02; medium size: $t$ (11) = 4.40, BH-adjusted $p$ = .003, Cohen's $d$ = 1.27; and large size: $t$ (11) = 3.06, BH-adjusted $p$ = .013, Cohen's $d$ = 0.88. The results indicate that there was the overestimation phenomenon in each area condition in either the 50- or 150-element condition.

The ANOVA and t-test indicated that the overestimation phenomenon occurred for all the numbers of elements used in each area condition; the amount depended on its size. As shown in the left panel of Fig 6, the perceived number of elements is larger for the small area than the medium or large area for the 150-element standard, while it remains constant for the 50-element standard. The function of the perceived number for 150 elements is similar to that for 100 elements (see Fig 5). This indicates that both the area containing elements and the number of elements play a role in overestimating numerosity.

As in Experiments 1–3, we show each observer's PSE ratio as a function of the area in the right two panels of Fig 6; the upper and lower panels refer to the 50- and 150-element conditions, respectively. In the panels, the same observer is connected by a line. The upper panel shows that every observer's PSE ratio was more than zero except for one, two, and three observers in the small, medium, and large conditions, respectively. The panel also shows that the PSE ratio was relatively constant as a function of the element size except for two observers whose PSE ratios increased as a function of the area. The lower panel shows that every observer's PSE ratio was more than zero except for one, two, and two observers in the small, medium, and large conditions, respectively. The panel also shows that the PSE ratio was higher in the medium condition for 11 out of 12 observers and in the large condition for 10 out of 12 observers than in the small condition. These results are consistent with the above conclusion that the area containing the elements affects the degree of the overestimation phenomenon for the standard with 150 elements; there is no effect for the standard with 50 elements.

The fact that the amount of 3-D overestimation depended on the element number and area of the standard stimulus may lead one to question if the density of the standard plays some role in the overestimation. The analysis of the density used in Experiments 3 and 4 suggests that it is less likely. For the standard with 50 elements, the amount of overestimation was relatively constant (17% on average) among the three areas (see Fig 5) where the density changes from 0.4 to 3.3 elements/arc deg$^2$. However, for the standard with 150 elements, the overestimation increased in the medium (28%) and large (30%) sizes, where the densities were 2.4 and 1.1 elements/arc deg$^2$, respectively; in the small size the increase was 12% where the density was 9.7 elements/arc deg$^2$. These results indicate that the overestimation did not depend on the density of the standard, at least not within the density range used.

The present results and those in Experiment 3 showed the effect of area on the amount of overestimation phenomenon when the element number was relatively large (100 and 150 elements) but not when it was relatively small (50 elements). The effect occurred when the area was more than 61.7 arc deg$^2$. These results are consistent with those in Aida [18], where the amount of overestimation was almost the same when the element number was varied from 12 to 60 for the area less than 37.2 arc deg$^2$. All these results are consistent with the idea that the area affects the amount of overestimation when the element number is relatively large but not when it is relatively small.

Interestingly, the effect of area has been reported for the 2-D elements as well as the 3-D elements in the literature. The effect was observed when the element number was relatively large [2, 12] or within its relatively wide range [8, 10] and was less likely to be observed when the number was relatively small [13]. We do not know whether the size effect observed for 2-D and 3-D stimuli can be attributed to a similar mechanism. However, our procedure to present a 2-D and a 3-D stimulus side-by-side suggests that the obtained effect for a 3-D stimulus may occur in a mechanism to mediate binocular depth perception.

## General discussion

This study conducted four experiments to test the overestimation of the number of 3-D elements compared to 2-D elements and examine which stimulus property of elements affects the overestimation. Experiment 1 showed that overestimation was relatively constant regardless of whether observers perceived the 3-D elements as overlapping surfaces or cylindrical volumes. Experiment 2 showed that the size of the elements did not affect the amount of overestimation within the size range. In Experiments 3 and 4, where the number of elements on a 3-D stimulus was 100 or 150, the overestimation amount was larger for the large area than for the small area. Moreover, in Experiment 4, the overestimation amount was constant when a 3-D

stimulus had 50 elements (note that in Experiments 1 and 2, the number of elements in a 3-D stimulus was 100). These results indicate that (a) the total-element overestimation phenomenon always occurred for all the conditions we used, (b) the area affected the amount of overestimation, while the overlapping stereo surface and size of elements did not, and (c) the effect of the area containing elements and the number of elements interact with the phenomenon.

In the literature, considerable research has reported that the mechanism to process binocular disparity defining the depth structure interacts with that to process visual primary properties: color (e.g., [28]), motion (e.g., [29]), spatial frequency (e.g., [30]), and density (e.g., [31]). Our finding that the number of elements appeared more numerous for a 3-D stimulus than a 2-D stimulus suggests that the mechanism to process numerosity may interact with that to process the binocular disparity. Accordingly, if numerosity is an independent primary visual property [32], the overestimation of 3-D elements [15–18] can be added to the list of the interaction between the mechanism to process binocular disparity and the visual primary properties. This claim is consistent with a recent finding that adaptation to a 3-D and 2-D stimulus containing the same elements resulted in different adaptation effects [33].

We discuss how the results can be explained by three previous models (back-surface bias, occlusion, and disparity-processing interference models). The back-surface bias model predicts that the overestimation phenomenon of a 3-D stimulus would occur only when the stimulus is perceived to have a back surface [15, 16]. The prediction agrees with most results except for Experiment 1, where the phenomenon was observed when the stimulus contained no background. Second, the occlusion model [16] assumes that the visual system adds extra elements when observers perceive overlapped stereo surfaces, and the number of unseen elements could grow with the size of the area [16]. The model is generally consistent with the results in Experiments 3 and 4 but needed help explaining the results of Experiment 1. Third, the disparity-processing interference model [17] assumes that processing the disparity of 3-D stimulus elements burdens the numerosity estimation and interferes with the normal functioning of the numerosity estimation process. The model generally agrees with all the results. However, we need to assume other mechanisms mediating the effects of the size of the area found in Experiment 3 and the number of elements found in Experiment 4.

Suppose no models can account for all the present and previous results, as discussed above and in the Introduction section. What form the final explanation will take is difficult to say. One possible explanation is that the three models should be complementary rather than competing. In particular, if a 3-D numerosity judgment mediates through a series of stages [34] where the processes assumed in the models could operate, most results would be explainable. However, we do not yet have a definitive answer for how the processes interact in a depth domain, suggesting that more research has to be conducted before we can fully understand 3-D overestimation in numerosity perception.

Considering the above, we speculate a process or processes for the area of a 3-D stimulus on the overestimation. As discussed in Experiment 4, the effect of the area may occur through a mechanism mediating depth perception. In the stereopsis literature, a second-order mechanism processing the contrast-based disparity information is proposed to depend on "the amount of disparity relative to stimulus width" (p. 2661) [35]. In the numerosity literature, Kramer et al. [36] found that observers could estimate numerosity from contrast-based motion information. This finding suggests that the second-order mechanism operates in numerosity estimation. Thus, if the processing load of the second-order disparity information increases with the size of the area, the effect of area on numerosity estimation obtained in Experiments 3 and 4 can be explained in general. However, this speculation does not explain that the effect was not obtained when the element size was small.

In this paragraph, we aim to understand why Bell et al. did not observe the overestimation phenomenon [20]. As discussed in the preamble of Experiment 2, the element size in Bell et al. [20] was slightly smaller than the smallest size in this experiment. However, we do not believe that such a slight difference in element size can fully explain the non-occurrence of the phenomenon. Instead, we speculate that the absence of the phenomenon can be attributed to the duration of a stimulus presentation across studies. While Aida et al. [16, 17], Aida [18], and our study presented the stereoscopic stimulus without a time limit, Schütz [15] presented it for 1.0 second, and Bell et al. [20] presented it for 0.5 seconds. According to Tam and Stelmach [26], the percentage of observers who can correctly detect a stereoscopic stimulus decreases as the presentation duration decreases. For example, when a stereoscopic stimulus was presented for 0.5 seconds, as in Bell [20], around 10% of the observers could not detect the stimuli. We therefore speculate that this limited presentation time might have affected the detectability of the stereoscopic stimulus and caused the results to imply no overestimation phenomenon in Bell et al. [20].

Finally, we note that the total-element overestimation phenomenon for a 3-D stimulus does not necessarily indicate that the perceived number of a 3-D stimulus is more than its physical number or that the 3-D numerosity estimation is inaccurate. In the literature, numerosity judgment for a 2-D stimulus is known to be underestimated (e.g., [8, 37–40]). For example, Krueger [39] found that the apparent numerosity of the elements presented on a flat surface (i.e., a 2-D stimulus) increased as a power function with exponents of approximately 0.81 when obtained with the magnitude estimation method. The finding indicates that the perceived number of elements is less than the physical number for the 2-D stimulus. However, as in this study and others [15–18], the number of 3-D elements is perceived as larger than 2-D elements when the two elements are presented side-by-side. If the number of elements in a 2-D stimulus is underestimated and that in the 3-D stimulus is overestimated when presented as in the present study, the perceived number of elements in a 3-D stimulus is not "overestimated" but could be closer to the physical number of elements presented. Thus, the total-element overestimation phenomenon does not necessarily mean that 3-D elements are overestimated in numerosity. Instead, the phenomenon may suggest that human observers can estimate the number of objects scattered in a 3-D scene more accurately than in a two-dimensional scene.

## Supporting information

**S1 File. Raw data for all experiments.**
(XLSX)

## Acknowledgments

Part of this study was reported in Vision Sciences Society (2017) [41]. We thank Yukiko Yoshida for conducting part of the experiments.

## Author Contributions

**Conceptualization:** Yusuke Matsuda, Saori Aida, Koichi Shimono.

**Data curation:** Yusuke Matsuda.

**Formal analysis:** Yusuke Matsuda.

**Funding acquisition:** Yusuke Matsuda, Saori Aida, Koichi Shimono.

**Investigation:** Yusuke Matsuda.

**Methodology:** Yusuke Matsuda.

**Project administration:** Yusuke Matsuda, Koichi Shimono.

**Resources:** Yusuke Matsuda, Koichi Shimono.

**Software:** Yusuke Matsuda.

**Supervision:** Yusuke Matsuda, Koichi Shimono.

**Validation:** Yusuke Matsuda.

**Visualization:** Yusuke Matsuda.

**Writing – original draft:** Yusuke Matsuda.

**Writing – review & editing:** Yusuke Matsuda, Saori Aida, Koichi Shimono.

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
