## [Decision Letter · Decision Letter 0]

24 Oct 2023

PONE-D-23-27033Effect of 3-D depth structure, element size, and area containing elements on total-element overestimation phenomenonPLOS ONE

Dear Dr. Matsuda,

Thank you for submitting your manuscript to PLOS ONE. After careful consideration, we feel that it has merit but does not fully meet PLOS ONE’s publication criteria as it currently stands. Therefore, we invite you to submit a revised version of the manuscript that addresses the points raised during the review process.

Both reviewers appreciated your efforts in revising your manuscript, and they raise only a few remaining points that need clarification. Addressing all reviewer comments should be straightforward, and I look forward to recieveing your revised manuscript. 

We look forward to receiving your revised manuscript.

Kind regards,

Guido Maiello

Academic Editor

PLOS ONE

Journal Requirements:

3. We noted in your submission details that a portion of your manuscript may have been presented or published elsewhere. [Some of the data from this experiment have been submitted to a conference and have been peer-reviewed and published as an abstract. Since the publication consisted only of a 274-words abstract, it did not constitute a dual submission from our point of view. We hope you agree. We added a PDF with the respective Abstract as additional information. The citation would be: Matsuda Y, Shimono K, Aida S. Overestimation of the number of elements in a three-dimensional stimulus is dependent on the size of the area containing the elements. J Vis 2017;17(10): 151. doi: 10.1167/17.10.151] Please clarify whether this [conference proceeding or publication] was peer-reviewed and formally published. If this work was previously peer-reviewed and published, in the cover letter please provide the reason that this work does not constitute dual publication and should be included in the current manuscript.

Reviewers' comments:

Reviewer's Responses to Questions

**Comments to the Author**

1. Is the manuscript technically sound, and do the data support the conclusions?

Reviewer #1: Yes

Reviewer #2: Yes

2. Has the statistical analysis been performed appropriately and rigorously? 

Reviewer #1: Yes

Reviewer #2: Yes

3. Have the authors made all data underlying the findings in their manuscript fully available?

Reviewer #1: Yes

Reviewer #2: Yes

4. Is the manuscript presented in an intelligible fashion and written in standard English?

Reviewer #1: Yes

Reviewer #2: Yes

5. Review Comments to the Author

Reviewer #1: In the last revision the authors wrote in their response letter that they conducted a baseline experiment for Experiment 1 -- 'The result of the baseline experiment showed that when 2-D stimuli were used as the standard and comparison, there was no numerosity overestimation for the standard stimulus'. Somehow I can't find the baseline experiment in the current manuscript -- it would be very helpful to include it in the paper. Other major questions are well answered.

Reviewer #2: The authors have addressed almost all of my concerns. The manuscript is streamlined and clear. I have a few outstanding minor queries and suggestions that I hope the authors will take onboard.

1. Experiment 2-4: the various combinations (3 main-variable manipulations x 7 numerosities x 2 standard locations x 10 repetitions) should lead to 420 trials, but is stated as 210. Is that a typo or were the locations not balanced (but were randomly chosen instead)?

2. A central motivating factor for the study has been the discrepancy between earlier results (Bell et al. and Aida et al. – in relation to observing overestimation in a 3-D volume). However, this discrepancy is not resolved by the current study, since element size does not seem to matter, and in the current experiments, overestimation is observed at all tested numerosities and areas. The authors speculate that the difference might lie in duration (line 439 – 455). Obviously testing it directly would be the ideal way forward. But if that cannot be done, please move this paragraph and discuss it in the General Discussion, rather than tuck it away in the results of Experiment 2.

3. My apologies for not being clearer in my earlier comment about the ‘control’ experiment. I was merely pointing out the inadequacy of the previous ‘baseline’ experiment to rule out response bias – I did not mean to confuse novelty and response bias. To really rule out response bias, the ‘standard’ 2-D stimulus should have been made distinct in some way. If you still find overestimation for the standard, it means that participants are biased towards an ‘interesting’ or ‘novel’ stimulus. Given that all 3-D stimulus in this study lead to overestimation, it is important to rule out such a bias towards 3-D stimulus. Yes, the authors make a case for why response bias might not be plausible here, but I don’t find it fully persuasive (see below).

a. The evidence for why response bias might have occurred can also be explained by response bias: The effect of disparity (stepwise 3-D) on overestimation can be explained by considering that the more depth that is observed, the more response bias there might be towards it. And perhaps it saturates at some point, hence there might not be much difference between clearly 3-D structures such as 2-POTS and 3-POTS with comparable disparities. In the current study, all manipulations lead to overestimation, indicating that such a bias cannot be ruled out. My suggestion was to do an experiment, with the same procedure, but instead of a 3-D structure, merely change the attributes of the standard stimulus elements. Thus, if participants have a bias towards a novel/interesting stimulus, they should pick that one too.

4. Line 785-787: Can the authors elaborate on why the disparity-processing model cannot account for the results of Experiments 3 and 4?

There still remain several typos and grammatical errors. Please go through carefully and fix them. Here, I point out a couple that might make a substantial difference in interpretation:

1. Abstract: Line 35 should be: ‘while the number of overlapping stereo surfaces.

2. Line 604: 195 x alpha might be missing a term: all the others are in the form of a+b*alpha.

6. PLOS authors have the option to publish the peer review history of their article (what does this mean?). If published, this will include your full peer review and any attached files.

Reviewer #1: No

Reviewer #2: No

---

## [Author Response · Author response to Decision Letter 0]

18 Dec 2023

Our responses to editor

Thank you for your comment.

In this revision, we addressed the points the editors pointed out.

1. Revised the manuscript based on PLOS ONE's style requirements.

3. Added details of previous conference presentations in the cover letter.

4. No need to update the information about 'Data Availability statement'.

5. Added 'ethics statement' to the 'Methods' section.

6. The reference list still needs to be updated in this revision.

Regarding your comment that "2. Financial Disclosure and Funding Information do not match," the information in 'Funding Information' is correct. Therefore, the 'Funding Information' has not been corrected in this resubmission. On the other hand, "Financial Disclosure" is not wrong, but it is misleading, so I would like to correct it. However, I cannot seem to correct "Financial Disclosure," so could you please change it to something like the following?

Financial Disclosure

This work was partly supported by JSPS (Japan Society for the Promotion of Science) KAKENHI (https://www.jsps.go.jp/english/e-grants/) Grant Numbers 19K20645(Grant-in-Aid for Young Scientists) for YM; 17K18187 (Grant-in-Aid for Young Scientists(B)) for SA; 21K18027 (Grant-in-Aid for Young Scientists) for SA; 23330215, 15H03463 (Grant-in-Aid for Scientific Research(B)) for KS. The funders had no role in study design, data collection and analysis, decision to publish, or preparation of manuscript.

Our responses to Reviewers

The following are our responses to Reviewer #1’s specific comments:

Reviewer #1: In the last revision the authors wrote in their response letter that they conducted a baseline experiment for Experiment 1 -- 'The result of the baseline experiment showed that when 2-D stimuli were used as the standard and comparison, there was no numerosity overestimation for the standard stimulus'. Somehow I can't find the baseline experiment in the current manuscript -- it would be very helpful to include it in the paper. Other major questions are well answered.

We apologize for not including our discussion regarding the baseline in the previous version of the manuscript. As requested, we have included this discussion in the revised manuscript. Please note that the baseline or control experiment was newly conducted. 

Reviewer #2: The authors have addressed almost all of my concerns. The manuscript is streamlined and clear. I have a few outstanding minor queries and suggestions that I hope the authors will take on board.

1. Experiment 2-4: the various combinations (3 main-variable manipulations x 7 numerosities x 2 standard locations x 10 repetitions) 

should lead to 420 trials, but is stated as 210. Is that a typo or were the locations not balanced (but were randomly chosen instead)?

We appreciate this comment. We decreased the trial number through Experiments 2-4, as compared with Experiment 1, in order to shorten the total time required to complete each experiment. The number of trials was 210, but the number of repetitions was 5—not 10.

2. A central motivating factor for the study has been the discrepancy between earlier results (Bell et al. and Aida et al. – in relation to observing overestimation in a 3-D volume). However, this discrepancy is not resolved by the current study, since element size does not seem to matter, and in the current experiments, overestimation is observed at all tested numerosities and areas. The authors speculate that the difference might lie in duration (line 439 – 455). Obviously testing it directly would be the ideal way forward. But if that cannot be done, please move this paragraph and discuss it in the General Discussion, rather than tuck it away in the results of Experiment 2.

We agree with this comment. We have moved our discussion on the discrepancy between the results of Bell et al. and Aida et al. to the General Discussion. 

3. My apologies for not being clearer in my earlier comment about the ‘control’ experiment. I was merely pointing out the inadequacy of the previous ‘baseline’ experiment to rule out response bias – I did not mean to confuse novelty and response bias. To really rule out response bias, the ‘standard’ 2-D stimulus should have been made distinct in some way. If you still find overestimation for the standard, it means that participants are biased towards an ‘interesting’ or ‘novel’ stimulus. Given that all 3-D stimulus in this study lead to overestimation, it is important to rule out such a bias towards 3-D stimulus. Yes, the authors make a case for why response bias might not be plausible here, but I don’t find it fully persuasive (see below).

a. The evidence for why response bias might have occurred can also be explained by response bias: The effect of disparity (stepwise 3-D) on overestimation can be explained by considering that the more depth that is observed, the more response bias there might be towards it. And perhaps it saturates at some point, hence there might not be much difference between clearly 3-D structures such as 2-POTS and 3-POTS with comparable disparities. In the current study, all manipulations lead to overestimation, indicating that such a bias cannot be ruled out. My suggestion was to do an experiment, with the same procedure, but instead of a 3-D structure, merely change the attributes of the standard stimulus elements. Thus, if participants have a bias towards a novel/interesting stimulus, they should pick that one too.

As per the reviewer’s suggestion, we conducted a control experiment by modifying one of the attributes of a standard stimulus. We maintained the same procedure and other stimulus characteristics in the control experiment as those in the main experiments. In this experiment, the standard and the comparison were 2-D stimuli that differed only in the color or shape of their elements. Specifically, the standard stimulus had triangular black or rectangular red elements, whereas the comparison stimulus had rectangular and black elements. The findings of the control experiment did not indicate the overestimation phenomenon. The result suggests that observers did not usually opt for a standard stimulus compared to the comparison stimulus.

4. Line 785-787: Can the authors elaborate on why the disparity-processing model cannot account for the results of Experiments 3 and 4?

We thank you for the comment. We have added our discussion regarding why the model cannot explain the results.

There still remain several typos and grammatical errors. Please go through carefully and fix them. Here, I point out a couple that might make a substantial difference in interpretation:

1. Abstract: Line 35 should be: ‘while the number of overlapping stereo surfaces.

2. Line 604: 195 x alpha might be missing a term: all the others are in the form of a+b*alpha.

We thank you for your careful reading. We have made the necessary corrections as suggested and have tried to fix our careless writing. We hope that we have dealt with all our typos. 

We hope you and the other two referees will find this revised version suitable for publication in PLOS ONE.

---

## [Decision Letter · Decision Letter 1]

18 Jan 2024

PONE-D-23-27033R1Effect of 3-D depth structure, element size, and area containing elements on total-element overestimation phenomenonPLOS ONE

Dear Dr. Matsuda,

Thank you for submitting your manuscript to PLOS ONE. After careful consideration, we feel that it has merit but does not fully meet PLOS ONE’s publication criteria as it currently stands. Therefore, we invite you to submit a revised version of the manuscript that addresses the points raised during the review process. The reviewer points out one minor inconsistency that should be corrected. Once this is done, I will accept the paper without need for further re-review. 

We look forward to receiving your revised manuscript.

Kind regards,

Guido Maiello

Academic Editor

PLOS ONE

Journal Requirements:

Reviewers' comments:

Reviewer's Responses to Questions

**Comments to the Author**

1. If the authors have adequately addressed your comments raised in a previous round of review and you feel that this manuscript is now acceptable for publication, you may indicate that here to bypass the “Comments to the Author” section, enter your conflict of interest statement in the “Confidential to Editor” section, and submit your "Accept" recommendation.

Reviewer #2: All comments have been addressed

2. Is the manuscript technically sound, and do the data support the conclusions?

Reviewer #2: Yes

3. Has the statistical analysis been performed appropriately and rigorously? 

Reviewer #2: Yes

4. Have the authors made all data underlying the findings in their manuscript fully available?

Reviewer #2: Yes

5. Is the manuscript presented in an intelligible fashion and written in standard English?

Reviewer #2: Yes

6. Review Comments to the Author

Reviewer #2: Thank you for conducting the control experiment. The results are convincing that there is no response bias towards one or the other stimulus. All my remaining concerns have been addressed.

One final thing. Thanks for clarifying that there are 5 repetitions per location in Experiments 2-4. However, in Experiments 3 and 4, it is still noted as 10 repetitions. See lines 521-522 ["there were 210 trials (three areas × seven numbers of elements × two locations of the standard × ten repetitions)"] and 642-644 [" there were 210 trials (three sizes of areas × seven numbers of elements × two locations of the standard × ten repetitions) for each observer. This is *mathematically impossible*. If you include two locations, then please change # of repetitions to 5, or omit locations and retain 10 repetitions. Otherwise, you get 420 trials (3x7x2x10 = 420).

7. PLOS authors have the option to publish the peer review history of their article (what does this mean?). If published, this will include your full peer review and any attached files.

Reviewer #2: No

---

## [Author Response · Author response to Decision Letter 1]

5 Feb 2024

The following are our responses to Reviewer #2’s specific comments:

Reviewer #2: Thank you for conducting the control experiment. The results are convincing that there is no response bias towards one or the other stimulus. All my remaining concerns have been addressed.

One final thing. Thanks for clarifying that there are 5 repetitions per location in Experiments 2-4. However, in Experiments 3 and 4, it is still noted as 10 repetitions. See lines 521-522 ["there were 210 trials (three areas × seven numbers of elements × two locations of the standard × ten repetitions)"] and 642-644 [" there were 210 trials (three sizes of areas × seven numbers of elements × two locations of the standard × ten repetitions) for each observer. This is *mathematically impossible*. If you include two locations, then please change # of repetitions to 5, or omit locations and retain 10 repetitions. Otherwise, you get 420 trials (3x7x2x10 = 420).

We apologize for the misleading revisions in our previous manuscript. In Experiments 1–4, the position of the standard stimulus was determined randomly in each trial rather than five times in each position. Therefore, there were a total of 210 trials in Experiments 3 and 4. The following text has been revised:

Line 297 – 299, 

The placement of the 3-D standard stimulus (to the left or right of the midsagittal plane for each trial and observer) was randomly determined.

Line 427 – 428,

The location of the 3-D standard was randomly determined, as in Experiment 1.

Line 521,

We removed the phrase “two locations of the standard.”

Line 521 – 522,

The location of the 3-D standard was randomly determined, as in Experiments 1 and 2.

Line 642,

We removed the phrase “two locations of the standard.”

Line 640 – 643,

The location of the 3-D standard was randomly determined using the same method as in Experiments 1–3. Similar to Experiments 1–3, the observer may have performed each session three times.

In addition, we revised grammatical problems, repeated expressions, typos, and other issues.

Line 239 – 242,

If the number of standard stimulus elements is 100 (which is the same as in Experiment 1), a PSE of 0.09 indicates that “the 100 elements of the standard stimulus and the 109 elements of the comparison stimulus are perceived as the same number.”

Line 271 – 272,

The distance between the center of the standard and comparison stimuli was 13.8 arc deg.

Line 414 – 415,

and the center-to-center distance between the standard and comparison stimuli was 13.8 arc deg.

Line 501 – 503,

The center-to-center distance between the standard and comparison stimuli was the same for all three size conditions, i.e., was 13.8 arc deg, as in Experiments 1 and 2.

Line 610 – 611,

The center-to-center distance between the standard and comparison stimuli was 13.8 arc deg in all three size conditions, as in Experiment 3.

Line 661

areas for the 50-element standard (adj.ps > .05). 

Other minor revised are highlighted in the manuscript.

We hope that you will find this revised version suitable for publication in PLOS ONE.

Yours sincerely, 

Yusuke Matsuda

Assistant Professor

Suwa University of Science

Department of applied Information Engineering

Faculty of Engineering

---

## [Editor Report · Decision Letter 2]

8 Feb 2024

Effect of 3-D depth structure, element size, and area containing elements on total-element overestimation phenomenon

PONE-D-23-27033R2

Dear Dr. Matsuda,

We’re pleased to inform you that your manuscript has been judged scientifically suitable for publication and will be formally accepted for publication once it meets all outstanding technical requirements.

Kind regards,

Guido Maiello

Academic Editor

PLOS ONE
---

## [Editor Report · Acceptance letter]

16 Feb 2024

PONE-D-23-27033R2 

PLOS ONE

Dear Dr. Matsuda, 

I'm pleased to inform you that your manuscript has been deemed suitable for publication in PLOS ONE. Congratulations! Your manuscript is now being handed over to our production team.

Kind regards, 

on behalf of

Dr. Guido Maiello 

Academic Editor

PLOS ONE